# Emergent communication in human-machine games

**Nicolo' Brandizzi, Luca Iocchi**

Department of Computer, Automation and Management Engineering, Sapienza University of Rome, via Ariosto 25 Roma 00185, Italy, IT

`{brandizzi,iocchi}@diag.uniroma1.it`

## ABSTRACT

In this paper, we present how Emergent Communication (EmeCom) literature can be used to characterize a subset of human communication ability using recurrent features. EmeCom does not directly require human involvement; however, we advocate that the entire field spawned from a desire to achieve human-machine interaction and that explicitly considering human-AI interaction is an important research direction. Our goal is to bring emergent communication researchers together by demonstrating that different approaches can be regarded as subsets of the same problem. Also, we recognize the importance of incorporating other aspects of human interaction into this field.

## 1 INTRODUCTION

Communication between humans and machines is regarded as a prerequisite to achieving artificial general intelligence (AGI); thus, researchers have been interested in developing AIs that can communicate with humans using natural language.

Recent technological advances and attention-based architectures Vaswani et al. (2017) made transformers the preferred solution for human-machine conversation, GPT-3 being the most prominent example Brown et al. (2020). Although transformers can achieve high performances in human communication systems, they learn through statistical inferences on pre-existing languages, which does not align with natural language emergence in humans. This training pipeline fails to capture the essence of communication which, in humans, emerged as a means to achieve a common goal rather than being the objective itself.

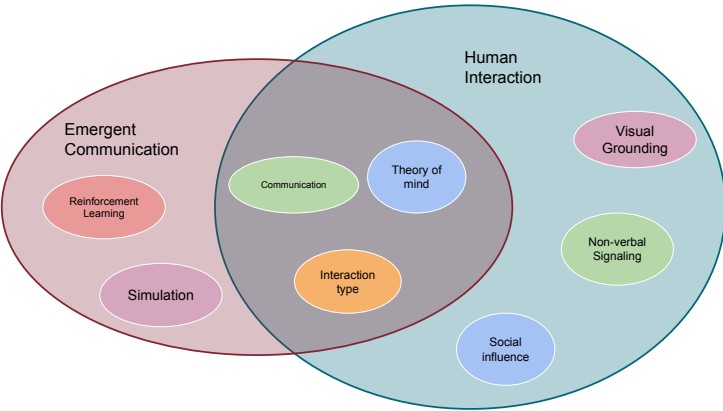

Figure 1: Framework division showing how different topics in EmeCom draw inspiration from a more general human interaction set.

This learning misalignment gave rise to the field of *Emergent Communication* (EmeCom) Wagner et al. (2003) to narrow the gap between machine and human language emergence. EmeCom aims to develop language through interaction between artificial agents, which can perceive the environment through observation and change its state by performing specific actions. Two fields heavily inspire this pipeline in artificial intelligence, Multi-Agent System (MAS) and Reinforcement Learning (RL) which we consider two fundamental pillars of human society and mind, respectively.

Games hold a meaningful impact on the learning process. Indeed, it has been shown how playing is a vital part of learning Spinka et al. (2001) that allows animals to develop the physical and psychological skills to handle unexpected events in which they experience a loss of control. Works such as Allee et al. (1949) emphasize how cheetahs cubs learn cooperation and social skills during playing, which will later be employed for hunts. Moreover, Tahmores (2011) identifies playing as fundamental for the proper development of social skills in children, and Kirriemuir & McFarlane (2004) investigates the literature on educational video games. Furthermore, RL finds its main application in games; thus, EmeCom is heavily based on game frameworks.

This paper aims to define common characteristics of EmeCom and highlight how each of them models a different aspect of human-human interactions. By specifying common characteristics, we can differentiate between several architectures and connect the literature based on our assumption.

**Paper Structure**   We focus on two main categories that we believe hold fundamental similarities with human interactions: *theory of mind*, Sec. 3, where agents are aware of other intelligent entities in the environment and actively try to model their behavior, and *interaction types*, Sec. 2, defining the possible configuration between agents in a shared environment. Due to space limitation, we omitted another fundamental aspect of EmeCom: the *game environment* itself, which can be divided in *communication-goal* [1] and *communication-aid* [2]. Finally, we provide a complete table of referenced papers, Table 1 with a correlated plot of papers published by year Fig. 2.

## 2   INTERACTION TYPE

In this section, we focus on the types of interaction among players. Although real-world interactions are complex, our goal is to split them into fixed categories and study the contribution of each one.

We must first introduce the difference between internal and external interaction types. Given a set $A_N$ of $N$ agents, we define a team $T_m \subseteq A_N$ as a subset of agents and we denote with $\{T_1, \dots T_k\}$ a partition of all the agents in $k$ disjoint teams with $\sum_{i=1,\dots,k} |T_i| = N$.

Given a partition of agents (i.e., a set of teams), we define a game to be cooperative when any negotiation between teams lead to a sub-optimal outcome $S_{so}$ for all the party involved. While each team actively tries to reach an optimal state $S_o$, trough negotiation it can avoid the worst outcome $S_p$ where $S_o < S_{so} < S_p$. On the other hand, a zero-sum game competitive setting always implies a worst outcome for all teams but one. In the latter, negotiation becomes ineffective, and each team must prevail over the others.

Now we can introduce the concept of *internal cooperation*, where all the agents of one team are cooperative with one another. This is a necessary condition for the emergence of communication to arise. Indeed, to the best of our knowledge, there are no works in EmeCom tackling internal competitiveness. On the other hand, *external cooperation* is defined as two or more disjoint teams of agents sharing a common goal, thus being incentivized to cooperate. Differently, *external competitiveness* arises in zero-sum games, where one team's victory implies a pessimal outcome for all other teams. In these cases, negotiation is not useful.

**Internal Cooperation**   As discussed, cooperativeness is a fundamental aspect for communication to emerge Smith (2010); Nowak & Krakauer (1999). As shown in Table 1, most of the treated works deal with internal cooperation settings. Most notably, Cao et al. (2018) studies how pro-social agents favor cheap talks and achieve better results than selfish ones. On the same line, Graesser

---

[1]Comprehending those games where communication is the objective itself, such as Referential games (Ref) and Task and Talk (TnT).

[2]Where communication assists the agents in achieving a goal

et al. (2019) suggests how intricate language evolution can emerge from simple social interactions between agents.

**External Cooperation**  External cooperation occurs when collaboration is extended between teams of agents and can be split into two types. The first one is the standard instance for external cooperation. Given a set of disjoint teams, the procedure is to initialize all the agents at the same time and let them interact with the environment, and each other Evtimova et al. (2018); Lowe et al. (2017). The peculiarity of this method is how every agent is initialized simultaneously and has no prior experience with the environment. Differently, when teams are pre-trained with one another before being introduced to other teams, this is referred to as population learning. This setting is heavily influenced by the study of human nature, specifically by language development between different cultures Briscoe (2002). In fact, the number of agents can influence the performance directly. It has been shown that a population of agents generalizes better than a pair. Moreover, the interaction between populations of agents can emerge a language that is easier to teach and understand Li & Bowling (2019); Lowe et al. (2019); Fitzgerald (2019).

**External Competitiveness**  On the other hand, Liang et al. (2020) show how competitiveness can develop better communication protocols and improve general performance. In particular, they speculate how competition among a set of agents leads to the development of a communication protocol that prioritizes compositionality, performance, and convergence. It is worth noting that Liang et al. work with cooperative groups of competitive agents, thus making this setting more akin to human experience. The first three categories deal with the spatial component of human connection. The number of agents/teams in the environment can be utilized to characterize the differences between them. On the other hand, humans live in a temporal dimension in which information and experiences are passed down from one generation to the next. This temporal aspect drives an extra interaction type known as iterative learning, which is opposed to previously addressed ones.

**Iterative Learning**  The concept of iterative learning is orthogonal to the population one. While this category does not strictly refer to an interaction type, we believe it captures a fundamental aspect of human interactions; thus, we decided to include it in this section. Indeed iterative learning is a cultural transmission where a population of agents passes their knowledge to a new one, and the loop repeats indefinitely. In the field of linguistics, iterative learning has been directly compared to a bottleneck in the learning system, which allows for generalization Nowak & Krakauer (1999). In Kirby et al. (2014) the authors summarize the approaches taken to study the emergence of natural language with an iterative point of view. They report various laboratory experiments in which iterative learning caused the shift towards languages that are consistent with prior biases. The same line of research was adopted by Scott-Phillips & Kirby (2010), where the author proposes a homonymy [3] filter to increase the structure and compositionality of the language through various generations.

On the same line, Ren et al. (2020) theorizes a correlation between compositionality, generalization, and topological similarity. For this reason, the authors build a training pipeline where artificial agents are trained on data generated by past versions of themselves and show how this advantage the emergence of language with high topological similarity.

These, and more works Dagan et al. (2021); Cogswell et al. (2019); Lu et al. (2020), show how language constructs passed down from generation to generation of learning agents lead to easier to learn languages. On the other hand, Guo et al. (2019) reports how iterative learning leads to the emergence of a compositional language; however, the results are highly dependent on input representation.

## 3  THEORY OF MIND

A fundamental propriety arising in the EmeCom field is directly derived from human behavior and can be seen as an attempt to model other agents' beliefs. Indeed, as humans, we tend to build a belief model of how other humans may react to certain stimuli and update it with each new observation Gopnik & Wellman (1992); Premack & Woodruff (1978). This behavior was first theorized in

---

[3]Relation between words that have the same form but different meanings (e.g., a pen: a small enclosure for animals; a female swan); common in natural languages.

Premack & Woodruff (1978) and named Theory of Mind (ToM), after the humans' ability to represent the mental states of others. More recently, Rabinowitz et al. (2018) applied ToM to let artificial agents build a model of other agents' observation and behavior alone.

The following paragraphs identify two main approaches to tackle the modeling problem: (i) *Agent's modeling*, where the artificial agents are aware of others in the game and actively model their behavior to some extent; (ii) *Influencing others*, a direct extension of agent's modeling, where agents also manipulate other agents' behavior based on their objective.

**Agents' Modeling.** Modeling other agents is a well-established line of work in the MARL community. Although modeling other humans as separated entities is natural for us, machines need specific formulations to approximate the same behavior.

When considering the original ToM, these formulations can leverage the similarities between agents' belief systems. In this case, a communication message can be generated by maximizing an agent's own understanding. This procedure takes the name of *obverter technique* and has shown effective for the emergence of compositional languages Choi et al. (2018); Bogin et al. (2018).

On the other hand, mental models can also be based on other agents' actions and perceptions without assuming similar belief systems. For example, Raileanu et al. (2018) augment the agents' policy with a prediction of other agents' behavior and show that the agents can learn better policies using their estimate of the other players' goals in both cooperative and competitive settings. However, this work considers environments where communication is not present.

Lowe et al. Lowe et al. (2017) describe agents adapting to each other when trained in conjunction; this finding led to study agents who can reason about other agents and adjust the communication protocol accordingly Andreas & Klein (2016). On the same line, Rodriguez et al. Rodriguez et al. (2019) describe agents modeling the conceptual understanding of other agents by switching partners with different proprieties; for example, a color-blinded agent will not benefit from a color-oriented description of an image. Moreover, Grover et al. Grover et al. (2018) show how to split the representation learning into two parts: a *generative* embedding simulates an agent's policy, while a *discriminative* one distinguishes one agent from the other. The combination of these two representations yields the best results in two physics simulation environments, especially when the learning takes place in an offline manner[4].

The above mentioned works show how agent modeling allows communication to quickly adapt and specialize to the task at hand. However, such specialization leads to languages that are difficult to interpret by humans.

In a multi-agent environment, whom to communicate with can be as important as the content of the message itself. For instance, Das et al. Das et al. (2019) consider a MAS setting and build an architecture, TarMac [5], based on soft-attention to let the agents choose their target. More precisely, each message is a continuous value split into two components: a value $v$ which encodes the information that an agent wants to send, and a signature $k$ multiplied with a query vector $q$. The dot product is normalized and used as a similarity measure between the sender signature and the receiver query. Intuitively, attention weights are high when both sender and receiver predict similar signature and query vectors. Following Lowe et al. (2017), the authors apply centralized critic training by temporal difference and decentralized execution with RMSProp. They report improved performances for their method, specifically how multiple rounds of communication lead to significantly higher performance than simply increasing message size. This finding opens a new horizon on the possibility to increase performances in a MAS since it does not require signal sharing for and to each agent in the system. Future research could also investigate the behavior of TarMac when agent communication is associated with a cost.

**Influencing Others.** The next natural step in this direction is given in Foerster et al. (2018) where the authors propose the Learning with Opponent Learning Awareness (LOLA) framework, which attempts to model the opponent's policy and actively tries to influence it using policy gradient optimization. This then evolves into higher-order LOLA, where agents are aware of opponents trying to

---

[4]They still report an increase winning chance with online representation learning compared to no representation at all.

[5]Targeted multi-agent communication

influence them, leading to third-order derivatives, which are computationally expensive. This same approach can be applied in mixed human-robot settings, such as in Xie et al. (2020), where the author build an RL environment where the agent employs an encoder-decoder architecture to model the action of a human being. This model is then used to approximate the human policy to maximize the total discounted reward. While most of the treated literature concerns MAS with artificial agent only, just a handful of papers consider mixed human-robot teams Jaques et al. (2019); Finn et al. (2017); Hawkins et al. (2020); Xie et al. (2020). As a matter of fact, we believe that this line of work should be investigated more deeply in the field and all the above mentioned paper could be extended in the same way.

## 4 CONCLUSION

In this paper, we carried out an analysis of the current emergent communication literature. We defined two main proprieties, namely **theory of mind**, Sec 3, and **interaction type**, Sec 2, and summarized the literature tracing parallelism with human real-world interactions. *Theory of mind* takes direct inspiration from the human ability to model other intelligent entities as individuals separated from the environment. A natural extension is *influencing other* agents based on the previous modeling.

On the other hand, we introduce the concept of *internal and external interactions*, which happen between agents belonging to the same and different teams, respectively. This allows us to define a new kind of *external competitiveness* setting, seen as an environment where only one team can win while the others experience the worst outcome. Finally, we offer a complete list of the treated works, Table 1, where we correlate papers with the relative category they fall in, and a plot showing the publication year in Fig 2.

Our work can assist interested researchers in approaching this new and exciting field and connect experts in the area under a common objective. In fact, we promote the idea that EmeCom has spawned by the necessity for human-robot collaboration; thus, the literature can be seen as tackling diverse aspects of human-to-human interaction. Under this assumption, we aim to connect apparently diverse works under a unified line of research.

**Future Work**   Most of the analyzed literature refers to multi-agent systems where the entities are artificial only; thus, the characteristics of human communication are approximated in a close setting. However, having a human-in-the-loop approach and, more explicitly, considering mixed human-robot teams is a promising research line. Finally, we would like to bring to attention other aspects of human interaction not typically discussed in EmeCom literature, such as visual grounding Antol et al. (2015) and nonverbal communication Breazeal et al. (2005).

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

| Work | Theory of mind | | Interaction Type | | | | Game Environment | | |
|---|---|---|---|---|---|---|---|---|---|
| | Modeling | Influencing | Int. Coop | Ext. Coop | Ext. Comp | Iter. | Com-goal | Com-aid | No-com |
| Jaques et al. (2019) | x | x | x | x | x | | Ref | x | |
| Andreas & Klein (2016) | x | x | x | | | | visual-Ref | | |
| Evtimova et al. (2018) | x | | x | | | | | | x |
| Foerster et al. (2018) | x | x | x | | | | Ref | | |
| Hawkins et al. (2020) | x | x | x | | | | Ref | | |
| Rodriguez et al. (2019) | x | | x | x | | | Ref | | |
| Xie et al. (2020) | x | x | x | | | | | | x |
| Rabinowitz et al. (2018) | x | | x | | | | | | |
| Brandizzi et al. (2021) | Trustworthiness | | x | x | | | | ww | |
| Cao et al. (2018) | Trustworthiness | | x | x | x | | | negotiation | |
| Raileanu et al. (2018) | Trustworthiness | | x | | | | | navigation | x |
| Das et al. (2019) | speaker-selection | x | x | x | x | | Ref | | |
| Choi et al. (2018) | Obverter | | x | | | | | x | |
| Bogin et al. (2018) | Obverter | | x | | | | x | | |
| Grover et al. (2018) | imitation-learning | | x | x | | | | navigation | |
| Bachrach et al. (2020) | | | x | x | | | | | x |
| Das et al. (2017) | | | x | | | | visual-TnT | | |
| Fitzgerald (2019) | | | x | x | | | Ref | | |
| Graesser et al. (2019) | | | x | x | | x | Multi-Ref | | |
| Kirby et al. (2014) | | | x | | | x | x | | |
| Kottur et al. (2017) | | | x | | | | TnT | | |
| Li & Bowling (2019) | | | x | x | | x | Ref | | |
| Liang et al. (2020) | | | x | | | x | TnT | | |
| Mordatch & Abbeel (2018) | | | x | | x | | | navigation | |
| Guo et al. (2019) | | | x | x | | x | Ref | x | |
| Ren et al. (2020) | | | x | | | x | | | |
| Cogswell et al. (2019) | | | x | x | | x | TnT | | |
| Dagan et al. (2021) | | | x | x | | x | Ref | | |

Table 1: Complete list of treated works and related categories.

**Theory of Mind** refers to Sec. 3, where instances are either classified as *modeling* (other agents) and *influencing*. Some rows also distinct between the kind of modeling embedded into agents: trustworthiness is investigated in competitive settings where the nature of a received message is not clear a priory; speaker-selection looks into the efficiency of communication; imitation learning is concerned with belief modeling based on non communicative actions; obverter refers to the obverter technique.

**Interaction type** includes *internal/external cooperation external competitiveness*, and *iterative learning*, as defined in Sec. 2. External includes multiple teams of interacting agents, while internal refers to the dynamic of a single team.

Lastly, **game environment**, categorizes the setting based on the role of communication which can be the goal of the game, *com-goal*, of assistance, *com-aid*, or not present, *no-com*. *Com-goal* can be either a referential game, Ref, a Task and Talk, TnT, or a visual by-product of the two. *Com-aid* mainly differentiate to negotiation or navigation tasks.

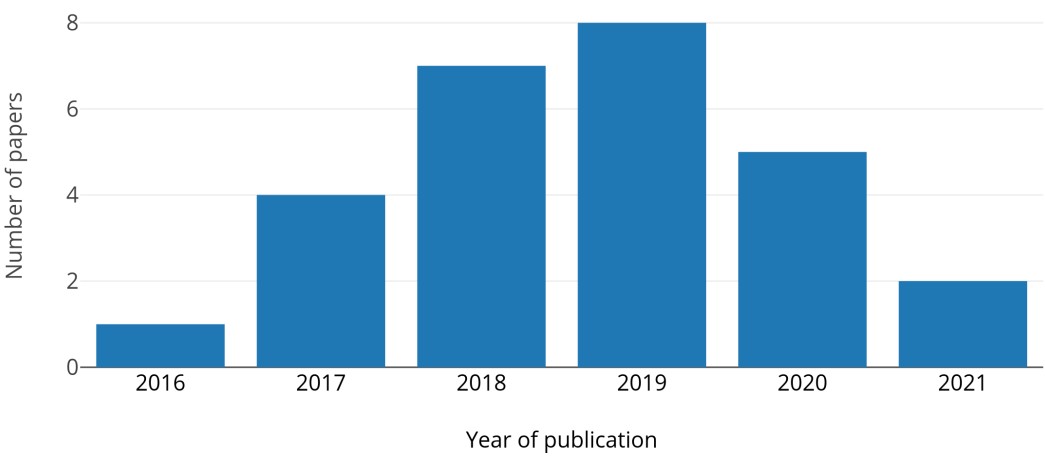

Figure 2: Number of papers per year of publication

