# OpenReview forum: "Emergent communication in human-machine games"
_ICLR.cc/2022/Workshop/EmeCom — EmeCom Workshop at ICLR 2022_

### Official Review · Reviewer_RNAN · 2022-03-22
**Relevant for the workshop but could be clearer and more precise**

**Rating:** Weak accept
**Confidence:** 4

**Review:**

**Summary**

This paper proposes an overview of different EC approaches and manage to show how they relate to common ideas. They propose to structure the literature by grouping recent works into two main categories: the type of interaction and modeling others.

**Main review**

This paper is relevant for this year workshop as it proposes different ways of structuring research in neural EC. Indeed, the neural EC field flourished a few years ago and it is timely to take a step back to define sub lines of research. First, this paper proposes to classify past works depending on the type of interactions involved in the simulations: internal/external cooperation/competition and iterated learning. This classification is meaningful as it corresponds to different types of human interactions and lead to different optimization. Second, the paper discusses how experiments can mimic the fact that humans manage to model others mind, and as an application, try to influence others.
Since these topics are relevant to discuss in the workshop, I recommend to **accept** the paper.

**Comments**

However, I think the paper could be clearer and more precise:

- In its form, I find the paper quite difficult to read. If I appreciate the idea of defining sub lines of research, I do not get the overall coherence of the paper. Each paragraph of the paper makes totally sense but I have difficulties to see how they contribute to an overall unique message. For example, I do not see why iterated learning is introduced and how it relates to previous discussions. More important, to me, Section 3 is orthogonal to Section 2 and I do not see a clear advantage of describing those two categories in the same paper, especially since each section could have been more detailed. All in all, I do not really get the main message of the paper and I do not see how the paper shows how EC projects could next connect under common lines of research.

- The paper is an overview of the literature but I find some lack of precision when introducing references, at least for the literature I am familiar with. For example, in the paragraph on Iterated Learning, it is surprising to see that no paper involving neural networks simulations are mentioned even though lots of neural EC works are inspired by IL (eg. [1,2,3]).  As another example, the paper mentions that generalization is improved with populations of agents but the impact of population is mitigated in the papers mentioned. Li & Bowling (2019) report no clear interest of using population (the paper mainly focused on reseting which is more related to IL). Fitzgerald (2019) displays an interest of having more than one pair of agents but shows that the impact of population is not clearly understood. Mitigated results linked to populations are also displayed in additional works [1,4,5,6].

In conclusion, there are a lot of interesting topics in the paper that could initiate relevant discussions. However,  the paper would have benefit from focusing on one of the two sections (Section 2 & 3) to ease reading, clear the position and improve the work on references, all the more each of the topics are very interesting.

**Minor remark**

- I am not a specialist but I think Theory of Mind and its applications to RL agents [7] would perfectly fit your Section on « modeling others »

**References**

- [1] Michael Cogswell, Jiasen Lu, Stefan Lee, Devi Parikh, and Dhruv Batra. Emergence of compositional language with deep generational transmission. arXiv preprint arXiv:1904.09067, 2019.

- [2] Yi Ren, Shangmin Guo, Matthieu Labeau, Shay B. Cohen, and Simon Kirby. Compositional languages emerge in a neural iterated learning model. In Proc. of International Conference on Learning Representations (ICLR), 2020.

- [3] Yuchen Lu, Soumye Singhal, Florian Strub, Aaron Courville, and Olivier Pietquin. Countering language drift with seeded iterated learning. In Proc. of International Conference on Machine Learning (ICML), 2020.

- [4] Olivier Tieleman, Angeliki Lazaridou, Shibl Mourad, Charles Blundell, and Doina Precup. Shaping representations through communication: community size effect in artificial learning systems. Visually Grounded Interaction and Language (ViGIL) Workshop, 2019.

- [5] Rahma Chaabouni, Florian Strub, Florent Altché, Eugene Tarassov, Corentin Tallec, Elnaz Davoodi, Kory Wallace Mathewson, Olivier Tieleman, Angeliki Lazaridou, Bilal Piot. Emergent communication at scale. International Conference on Learning Representations (ICLR) 2022

- [6] Mathieu Rita, Florian Strub, Jean-Bastien Grill, Olivier Pietquin, Emmanuel Dupoux. On the role of population heterogeneity in emergent communication. International Conference on Learning Representations (ICLR) 2022

- [7] Neil Rabinowitz, Frank Perbet, Francis Song, Chiyuan Zhang, S. M. Ali Eslami, Matthew Botvinick. Machine Theory of Mind. Proceedings of the 35th International Conference on Machine Learning, PMLR 80:4218-4227, 2018.

---

### Official Review · Reviewer_fmmE · 2022-03-24

**Rating:** Weak accept
**Confidence:** 3

**Review:**

Summary of the contributions:

This paper proposes to review some literature of Emergent Communication (EC) in order to highlight bridges both in-between studies focusing on artificial agents, and also with the ones focusing on human subjects. Three main dimensions/categories are identified, towit, features of the ‘game environment’, agent’s ‘interaction types’, and the extent to which abilities to ‘model others’
are accounted for.

This paper discusses and further refines the last two dimensions, and it eventually presents (in appendix) a table that breaks down the discussed works along all dimensions.

Strong of having identified shared interests and research directions, the paper remarks that research directions falling under the umbrella of human-in-the-loop approach (as opposed to approximating/simulating interactions with humans) have been kept relatively untouched and, thus, they should be prioritised to push the field forward.


Novelty, relevance, significance:

Given the goal of this year’s workshop to foster inter-disciplinary discussions, I think that this paper is very relevant as it highlights inter(and intra)-disciplinary bridges. Nevertheless, in its current form, it seems uncomplete to me as some relevant pieces of literature are missing:

•iterated learning: Guo et al. [2019], Ren et al. [2020], Dagan et al. [2020],

•theory of mind/modeling others: obverter approach[Choi et al., 2018, Bogin et al., 2018, Korbak et al., 2019] and (social) influence[Lowe et al.,2019, Jaques et al., 2018, Eccles et al., 2019].


Quality of writing/presentation:

I find the paper mainly well-organised, although it can be at times difficult to read. I would bring the authors attention onto the following main points:

•In Section 2, paragraphs 2, 3, and 4 seem disconnected from the rest of the section. Moreover, the definitions of ‘internal and external interaction types’ could use a more explicit reformulation, I feel.

•In Section 2, in the ‘Iterated Learning’ paragraph, on top of the addition of the previously mentionned missing literature, I think that the final mention regarding selfplay might need to be further fleshed out and given a proper paragraph for it to properly shine. I would propose referring to the following pieces of literature: Lu et al. [2020], Lowe et al. [2020]


Decision:

Given the potential of this paper to foster great discussion, I would be tempting to evaluate it as an ‘accept’ provided the proposed changes were addressed. Nevertheless, as it stands, the paper in incomplete and can be difficult to read, thus I am recommending a ‘weak accept’.


References:

B. Bogin, M. Geva, and J. Berant. Emergence of Communication in an Interactive World with Consistent Speakers. sep 2018. URL
http://arxiv.org/abs/1809.00549.

E. Choi, A. Lazaridou, and N. de Freitas. Compositional Obverter Communication Learning From Raw Visual Input. apr 2018. URL
http://arxiv.org/abs/1804.02341.

G. Dagan, D. Hupkes, and E. Bruni. Co-evolution of language and agents in referential games. Jan. 2020.

T. Eccles, Y. Bachrach, G. Lever, A. Lazaridou, and T. Graepel. Biases for emergent communication in multi-agent reinforcement learning. Dec. 2019.

S. Guo, Y. Ren, S. Havrylov, S. Frank, I. Titov, and K. Smith. The emergence of compositional languages for numeric concepts through iterated learning in neural agents. arXiv preprint arXiv:1910.05291, 2019.

N. Jaques, A. Lazaridou, E. Hughes, C. Gulcehre, P. A. Ortega, D. Strouse, J. Z. Leibo, and N. De Freitas. Social influence as intrinsic motivation for multi-agent deep reinforcement learning. arXiv preprint arXiv:1810.08647, 2018.

T. Korbak, J. Zubek,  L. Kuci ́nski, P. Mi lo ́s, and J. R ̧aczaszek-Leonardi. Developmentally motivated emergence of compositional communication via template transfer. oct 2019. URL http://arxiv.org/abs/1910.06079.

R. Lowe, J. Foerster, Y.-L. Boureau, J. Pineau, and Y. Dauphin. On the Pitfalls of Measuring Emergent Communication. mar 2019. URL
http://arxiv.org/abs/1903.05168.

R. Lowe, A. Gupta, J. Foerster, D. Kiela, and J. Pineau. On the interaction between supervision and self-play in emergent communication. Feb. 2020.

Y. Lu, S. Singhal, F. Strub, O. Pietquin, and A. Courville. Supervised seeded iterated learning for interactive language learning. Oct. 2020.

Y. Ren, S. Guo, M. Labeau, S. B. Cohen, and S. Kirby. Compositional Languages Emerge in a Neural Iterated Learning Model. feb 2020. URL http://arxiv.org/abs/2002.01365.

---

### Decision · Program_Chairs · 2022-03-25

**Decision:**

Accept

**Comment:**

This is an interesting position paper arguing for human-in-the-loop EC as an underexplored direction. We believe it will be a great addition to the workshop and thank the reviewers for their great suggestions and citations. We hope the authors can incorporate some of these into future work!